# The Use of Medicinal Plants in Blood Vessel Diseases: The Influence of Gender

**DOI:** 10.3390/life13040866

**Published:** 2023-03-23

**Authors:** Guglielmina Froldi

**Affiliations:** Department of Pharmaceutical and Pharmacological Sciences, University of Padova, 35131 Padova, Italy; g.froldi@unipd.it; Tel.: +39-049-827-5092; Fax: +39-049-827-5093

**Keywords:** vascular dysfunction, endothelium, plant extracts, gender, safety, botanicals, garlic

## Abstract

Data available in the literature on the use of herbal products to treat inflammation-related vascular diseases were considered in this study, while also assessing the influence of gender. To this end, the articles published in PubMed over the past 10 years that described the use of plant extracts in randomized clinical trials studying the effectiveness in vascular pathologies were analyzed. The difference in efficacy of plant-derived preparations in female and male subjects was always considered when reporting. The safety profiles of the selected plants were described, reporting unwanted effects in humans and also by searching the WHO database (VigiBase^®^). The medicinal plants considered were *Allium sativum*, *Campomanesia xanthocarpa*, *Sechium edule*, *Terminalia chebula*. Additionally, an innovative type of preparation consisting of plant-derived nanovesicles was also reported.

## 1. Introduction

Inflammation is a well-known biological response of the organism to chemical and physical injuries that help the healing of tissues [1]. However, the inflammatory reaction can be harmful when it is excessively great, causing acute organ failure, or when too persistent, triggering chronic systemic inflammation [2]. In general, the inflammatory status is manifested by an increase in serum C-reactive protein (CRP) level and the release of pro-inflammatory cytokines, such as interleukin-6 (IL-6) and tumor necrosis factor-α (TNF-α) [3]. Arterial and venous vessels are directly involved in inflammatory progression with alterations in endothelial permeability, causing liquid and protein leakage and tissue edema formation. TNF-α increases intracellular calcium and regulates myosin light chain kinase and RhoA, which disrupts endothelial junctions, reducing barrier function and enabling leukocyte transmigration [4,5]. Subjects with high levels of homocysteine, cholesterol, and triglycerides show a greater risk of stroke by approximately 50% than those with normal values [6]. Furthermore, the correction of hyperhomocysteinemia involves a reduction in stroke risk from 34% to 70% [7]. Among the factors that affect the amount of homocysteine present in the blood, are various physiological factors, such as age, sex, and body mass index [8,9]. In fact, women generally have lower levels than men [10]; therefore, homocysteine may be a possible factor responsible for gender differences in atherosclerosis and coronary artery disease [11].

In the blood vessel, acute and chronic inflammation results in endothelial dysfunction and vascular remodeling, affecting cardiovascular function. Several endogenous modulators play a role in vascular inflammatory processes; in particular, an increase in advanced glycation end-products (AGEs) causes activation of inflammatory pathways, oxidative stress, and procoagulant activity, leading to endothelial dysfunction [12,13]. Fishman et al. suggested that AGEs and their receptors may be useful biomarkers of the presence and severity of coronary artery disease [12]. Other inflammatory modulators are proinflammatory cytokines, such as TNF-α and interleukins (ILs) [14]. IL-1α and IL-1β are responsible for the disruption effects on the endothelial barrier, while nuclear factor kappa-light-chain-enhancer of activated B cells (NF-kB) and the mitogen-activated protein kinase (MAPK) signaling cause pro-atherogenic effects [15]. Moreover, the vascular adhesion protein-1 (VAP-1), also known as amine oxidase copper-containing 3 (AOC3), is a pro-inflammatory modulator with a greater expression in endothelial cells during inflammation [16,17]. VAP-1 is an ectoenzyme that catalyzes the oxidative deamination of primary amines and produces hydrogen peroxide, ammonium, and aldehydes, regulates leukocyte extravasation, and causes vascular damage and atherosclerosis [16,17]. Furthermore, endothelial dysfunction is mediated by the activation of enzymes such as heparinase and metalloproteinases (MMPs), which cleave glycoproteins anchored to the endothelial glycocalyx, which are activated by pro-inflammatory cytokines and reactive oxygen species (ROS) [18]. Furthermore, monosodium urate and cholesterol crystals, and islet amyloid polypeptides can damage the phagolysosome membrane and promote persistent activation of the NLRP3 inflammasome, also known as NALP3 or cryopyrin, causing severe inflammatory diseases such as gout, atherosclerosis, and type 2 diabetes mellitus [19,20,21]. Recently, extracellular vehicles such as platelet and endothelial microparticles have been reported to be involved in vascular regulation, including inflammatory and thrombotic homeostasis [22,23]. Therefore, vascular inflammation is clearly a complex condition that has several modulators.

The literature findings suggest that the treatment of inflammation with targeted drugs may promote the regression of vascular dysfunction [4,24,25]. However, non-steroid and steroid anti-inflammatory drugs have not shown protective effects against arterial stiffening; however, some promising results have been obtained with the use of selective inhibitors of IL-1β, IL-6, and TNF-α [4,25]. Unfortunately, the use of these inhibitors can cause the appearance of significant side effects, and their risk/benefit ratio remains to be further determined. In this context, plant-derived products with potential activity in the treatment of vascular dysfunction may be of interest. For this purpose, this study focused on the literature delineating potential plant products of utility in the treatment of cardiovascular diseases related to inflammatory damage. Particular attention was paid to the identification of differences in response to herbal medicines in women versus men.

## 2. Sex Differences in Vascular Function

Few studies considered the sex difference related to inflammation and metabolic and cardiovascular diseases. The general opinion is that men tend to have a worse risk factor profile than women, although this relationship changes with advancing age [26]. Differences in the prevalence of cardiovascular diseases were observed in premenopausal women compared to men of the same age; clinical data suggest that women are protected against various cardiovascular diseases primarily during the fertile period of life [27,28,29,30]. In fact, estrogens increase vascular NO^•^ signaling, improving vasodilatation and insulin responsiveness, protecting against diabetes mellitus [30]. However, the relative risk of cardiovascular disease morbidity and mortality in subjects affected by diabetes mellitus ranges from 1 to 3 in men and from 2 to 5 in women, related to those without diabetes [31]. Furthermore, clinical studies failed to demonstrate that hormone replacement therapy (HRT) in postmenopausal women could improve cardiovascular outcomes [30]. It can also be observed that men and women have different lifestyle risk factors, such as male behavior that may include more frequently cigarette smoking, alcohol abuse, higher intake of red meat, and lower fruit and vegetable consumption [32]. Effectively, in part, behavior differences could explain why women live longer than men [33]. Focus has been on the relationship between the Western lifestyle and chronic metabolic inflammatory diseases and also on finding preventive approaches [34]. A meta-analysis that considered a total of 462,194 participants demonstrated that high ingestion of fruits and vegetables (flavonoids) was inversely associated with the risk of total cardiovascular disease mortality [35]. Recently, Parmenter et al. (2022) found that in older women, a higher intake of habitual dietary flavonoids, mainly black tea, is associated with less extensive abdominal aortic calcification [36]. Previous authors have also shown that high black tea consumption is associated with lower coronary artery and abdominal aorta calcifications [37,38,39].

## 3. Materials and Methods

This study considered articles identified using a PubMed search strategy up to January 2023. The terms included in the search query were ‘blood vessel’, ‘inflammation’, and ‘plant-derived compound’. The inclusion criteria for the search were as follows: 1. Published in the last 10 years, 2. Full texts, 3. English language. Of a total of 16,395 articles included, some were excluded according to other inclusion criteria, such as ‘endothelium’ and ‘female’, excluding the term ‘review’ (Appendix A). A total of 20 articles was reviewed according to their relevance to the selected topic. Additionally, other references were examined also through Google Scholar to find additional relevant articles, including in vitro and in vivo studies to facilitate the explanation of the pharmacological mechanisms. The safety profiles of the selected plants were described, along with the unwanted effects reported in humans, also by searching VigiBase^®^, the WHO global database of possible side effects of medicinal products [40].

## 4. Results

The selected medicinal plants evaluated in this review were *Allium sativum*, *Campomanesia xanthocarpa*, *Sechium edule*, and *Terminalia chebula*. Additionally, an innovative type of preparation consisting of plant-derived nanovesicles was considered. Table 1 reports a synthesis of the clinical trials that detected the efficacy of the medicinal plants considered in this review.

### 4.1. Allium sativum L.

#### 4.1.1. Botanical Characteristics

Garlic (Amaryllidaceae) is a herbaceous plant whose bulbs are often used to flavor different types of food, widely known for its typical smell and taste [41]. Its alimentary use is widespread throughout the world, and its medicinal use is well-known in popular medicine as crude drug, standardized extracts, and also as a food supplement [42]. The bulbs are harvested in late spring and early summer, then dried in the shade at 40 °C, to enable storage [41]. Several traditional uses are known, such as antimicrobial, diuretic, vermifuge, adjuvant in the prevention of atherosclerosis and the relief of the common cold [43,44].

#### 4.1.2. Phytoconstituents and Preclinical Activity

Among the most peculiar constituents, there are several sulfur compounds, including alliin, an odorless compound, which in turn is transformed by the alliinase enzyme into allicin that has the typical garlic smell [45,46,47,48]. Allicin is a diallyl thiosulfinate considered one of the most active components, although other sulfur compounds provide garlic properties, such as ajoene, allyl propyl disulfide, diallyl trisulfide, and S-allylcysteine [49]. Additionally, garlic contains saponins, flavonoids, vitamins, and minerals [41,50]. Numerous in vitro and in vivo studies suggested its efficacy in several human diseases [51,52]. Garlic powder, aged garlic, and garlic oil have shown antiplatelet and anticoagulant effects by interfering with cyclooxygenase-mediated thromboxane synthesis [49,53]. Garlic extracts showed antioxidant property, decreased expression of vascular endothelial growth factor (VEGF), hypoxia-inducible factor 1 alpha (HIF-1α), inducible nitric oxide synthase (iNOS), and metalloproteinase (MMP)-9 [54,55,56]. Furthermore, garlic prevents the expression of inflammatory cytokines such as IL-6 and monocyte chemoattractant protein-1 (MCP-1) in lipopolysaccharide (LPS)-stimulated 3T3-L1 adipocytes [57].
life-13-00866-t001_Table 1Table 1Effectiveness of herbal preparations used for the treatment of vascular diseases administered orally to human subjects.Natural ProductsClinical TrialsParticipantsDosageOutcomesRefs.*Allium sativum*(garlic)R, single-blind, PCR, DB, controlledR, DB, PCR, DB, PC50 pregnant subjects44 pregnant subjects92 obese subjects91 T2DM subjects8 weeks 800 mg day garlic (1 mg allicin)9 weeks 400 mg day garlic (1 mg allicin)3 months 400 mg day garlic extract (2% allicin)4 weeks 500 mg twice day garlic (2–3 mg allicin)Reduces systolic blood pressure, total cholesterolReduces CRP, increases GSHDecreases CRP, PAI-1, TG and LDL-C. Increases TACImproves visual acuity[46,58,59,60,61,62]*Campomanesia xanthocarpa*(guavirova)R, DB, PCR, DB, PCR, DB, pilot33 hypercholesterolemic subjects156 hypercholesterolemic subjects23 healthy subjects3 months 250 or 500 mg day dried leaves3 months 500, 750 and 1000 mg day dried leaves1000 mg day dried leavesDecreases TG, LDLDecreases TG, LDL, CRP, oxidative stress. Increases NO^•^Antiplatelet activity[63,64,65]*Sechium edule*(chayote)NDNDNDNDND*Terminalia chebula*(black myrobalan)R, DB, PCR, DB, PC56 subjects with metabolic syndrome60 T2DM subjects12 weeks 250 or 500 mg twice daily aqueous fruit extract12 weeks 250 or 500 mg twice daily aqueous fruit extracta. and b. Improves endothelial function, increases NO^•^, GSH, HDL, decreases CRP, HbA1c, MDA, TG, LDL, VLDL[66,67,68]Plant-derived nanovesiclesOpen-label20 healthy subjects3 months 1000 mg day *Citrus limon* EVsDecreases waist circumference in women[69]CRP: C-reactive protein; DB: double-blind; EVs: extracellular vesicles; GSH: glutathione; PC: placebo-controlled; T2DM: type 2 diabetes mellitus; HbA1c: glycosylated hemoglobin A1c; HDL: high-density lipoproteins; LDL-C, low density lipoprotein cholesterol; MDA: malondialdehyde; PAI-1: plasminogen activator inhibitor 1; PC: protein carbonyls; R: randomized trial; TG, triglycerides; TAC: total antioxidant capacity; TC, total cholesterol; HDL-C, high density lipoprotein cholesterol. ND: no documented clinical trials.


#### 4.1.3. Therapeutic Efficacy: Clinical Trials

Garlic is a very well-known plant that is used around the world. Studies on garlic preparations have mainly tested hypocholesterolemic, antihypertensive, antimicrobial, and, also, antitumor activities [54,55,56]. A randomized placebo-controlled trial (RCT) enrolled a total of 100 pregnant women at high risk of pre-eclampsia who were treated during the third trimester of pregnancy for 8 weeks with 800 mg garlic tablets (dried garlic powder containing 1 mg allicin, and ajoene) per day or placebo [60]. The treatment prevented the increase of total cholesterol and reduced hypertension [60]. Another RCT conducted in 44 pregnant women treated with 400 mg garlic tablet (equal to 400 mg garlic and 1 mg allicin) for 9 weeks showed reduced serum CRP levels and increased plasma glutathione in the treated women compared to the untreated group [61]. Another study showed that garlic supplementation (400 mg/day, garlic extract 2% allicin) positively modifies endothelial biomarkers of cardiovascular risk, suggesting that treatment can reduce chronic inflammation in obese individuals of both sexes [58]. Furthermore, one trial also suggested that 500 mg granulated garlic powder (2–3 mg allicin) can be considered as an adjuvant treatment in patients with diabetic macular edema [62]. Recently, the acute efficacy of 180 mg fermented garlic extract enriched with 7 mg inorganic nitrite (NO_2_^−^) in healthy women and men showed a significant decrease in both systolic and diastolic pressure 30 min after ingestion of the product [70].

Systematic reviews and meta-analysis evaluating the effects of garlic supplementation have generally reported an improvement of lipid profile and insulin-resistance; however, the low quality of the trials does not permit at the moment the real effectiveness of garlic preparations to be defined [71,72]. Overall, the data available in the literature support the use of garlic extracts in the treatment of vascular diseases mainly with an atherosclerotic basis, without apparent differences between the two sexes. However, additional RCTs with standardized extracts are required to confirm the therapeutic use of garlic in cardiovascular diseases.

#### 4.1.4. Safety

According to toxicity data from experimental studies and medicinal use, garlic preparations are generally considered safe in the usual dosage regimen. Female and male rats treated with 300 and 600 mg/kg day of an aqueous garlic bulb extract for 21 days showed changes in weight growth, biological parameters, and histological structures [73]. In humans, common side effects included odor and skin rash, and gastrointestinal upset [60,74,75]. In VigiBase^®^
*Allium sativum* has 232 reports of potential side effects from all countries, mainly Europe (31%), in patients of both sexes (female 49%, male 47%, unknown 5%), in adults and older people [40]. Figure 1 shows the types and percentages of reported side effects. Among these, there are mainly gastrointestinal disorders (18%, such as vomiting, gastrointestinal pain, diarrhea, etc.), general disorders (13%, such as drug interaction, asthenis, etc.), nervous system disorders (11%, such as dizziness, hemorrhages, skin disorders, etc.) and various others (Figure 1). The large number of side effects related to garlic use may depend on the very high consumption of this plant worldwide. In general, the use of garlic even for curative purposes is considered safe in humans, avoiding use in atopic subjects [51,76,77]. The medicinal use of garlic is not recommended during pregnancy and breastfeeding due to the absence of clinical evidence showing both efficacy and non-toxicity [44]. Furthermore, it is also not recommended in patients being treated with antiplatelet and anticoagulant drugs due to the increased risk of bleeding [44,75].

#### 4.1.5. Future Needs

The use of garlic preparations is widespread both in the diet and for healing purposes. To optimize its medical use, it is essential to define the type of formulation and the titer of sulfur derivatives, the dose, and the mode of administration, to standardize treatments and compare data from different clinical trials. Certainly, double-blind randomized clinical trials, with a sufficient number of subjects of both sexes are needed, to validate the use of garlic in therapy in the treatment of vascular diseases. Otherwise, unfortunately, it would remain only a traditional use, without valid evidence of effectiveness, the use on the basis of scientific evidence being renounced.

### 4.2. Campomanesia xanthocarpa Berg.

#### 4.2.1. Botanical Characteristics

*Campomanesia xanthocarpa* (Myrtaceae) is a semi-deciduous tree, commonly known as “guavirova”, which grows in Brazil, Argentina, Paraguay, and Uruguay. It has edible fruits with a succulent pulp and a sweet flavor [78]. Leaves are traditionally used in herbal teas to treat inflammatory, urinary, rheumatic diseases, high blood pressure, and high cholesterol [78,79].

#### 4.2.2. Phytoconstituents and Preclinical Activity

Leaves contain phenolic compounds, such as chlorogenic, gallic, ellagic and rosmarinic acids, glycosylated flavanols mainly of quercetin and myricetin, and pro-anthocyanidins [80,81,82,83,84]. Alkaloid theobromine (3,7-dimethyl-xanthine) was also identified in an aqueous infusion of leaves [83]. Acute administration of an aqueous extract showed a dose-dependent hypotensive effect in rats, by inhibiting the renin–angiotensin system through the block of the angiotensin II type 1 receptor (AT1R) and of calcium currents, as well as by K_ATP_ channel activation [83,84]. Furthermore, several studies reported the antioxidant activity of the fruits and leaves [80,82,85].

#### 4.2.3. Therapeutic Efficacy: Clinical Trials

The authors studied *Campomanesia xanthocarpa* on inflammatory processes, oxidative stress, and lipid biomarkers of hypercholesterolemia, showing a decrease in total cholesterol and LDL levels in treated hypercholesterolemic subjects [63,64]. A small trial in 33 hypercholesterolemic subjects treated with 250 and 500 mg capsules that contained dried *Campomanesia xanthocarpa* leaves for 90 days revealed a significant reduction in total cholesterol and LDL levels in hypercholesterolemic subjects with total cholesterol >240 mg/dL (*n* = 22) [63]. In addition, another trial involving a larger number of subjects treated for 90 days with 500, 750, and 1000 mg of dried encapsulated leaves also demonstrated anticholesterolemic activity [64]. These authors also suggested that this treatment attenuates oxidative stress and pro-inflammatory reactions, improving blood flow and endothelial function [64]. Furthermore, healthy subjects were treated with 1000 mg of powdered *Campomanesia xanthocarpa* leaves (*n* = 8) and compared to those treated with 100 mg of acetylsalicylic acid (ASA, *n* = 7), or 500 mg of *Campomanesia xanthocarpa* plus 50 mg of ASA (*n* = 7). The authors showed that *Campomanesia xanthocarpa* leaves have antiplatelet activity when administered at 1000 mg for 5 days alone, or at 500 mg with low doses of ASA [65].

#### 4.2.4. Safety

The extract of *Campomanesia xanthocarpa* leaf administered to male rats at 300 mg/kg iv caused cardiac depression with a dramatic drop in blood pressure and animal death [84]. In contrast, a 5000 mg/kg ethanol leaf extract administered orally to five male and five female mice did not show toxicity [81]. Genotoxic effects were observed after treatment with an aqueous leaf extract administered to male rats at 1000 mg/kg [82]; this observation should be taken into account in future studies. No adverse reactions were reported for the use of *Campomanesia xanthocarpa* (guavirova) in VigiBase^®^ [40]. No information is available in the literature on its safety during pregnancy or breastfeeding in humans.

#### 4.2.5. Future Needs

Preparations with dry leaves of *Campomanesia xanthocarpa* seem to have interesting hypocholesterolemic and antiplatelet activities, potentially useful in vascular diseases. However, no specific active compounds were identified in the formulations administered to the subjects, and therefore there is no reference compound for titration. Data from clinical trials are very limited, and there is certainly a need for double-blind randomized clinical trials with a sufficient number of subjects of both sexes to define the clinical usefulness of this medicinal plant.

### 4.3. Sechium edule (Jacq.) Sw.

#### 4.3.1. Botanical Characteristics

*Sechium edule* (Cucurbitaceae) is a perennial herbaceous climbing plant cultivated mainly by Asian and Latino-American populations for food use, in particular for Chayote fruits [86,87,88]. Likewise, fruits, roots, and leaves are known in traditional medicine against kidney stones, as a diuretic and antihypertensive [89,90,91,92,93]. Furthermore, alcoholic extracts showed a very good antimicrobial efficacy against all strains of multi-resistant Staphylococci and Enterococci [94]. Moreover, various leaf and seed preparations have shown remarkable antioxidant activity [95]. Several investigations in different animal models, such as rats, mice, and dogs, defined the capacity of this plant to reduce blood pressure [86,89,91,96]. The hydroalcoholic extract and the acetone fraction obtained from the roots of *Sechium edule* showed antihypertensive activity by a relaxant effect on blood vessels [91,97].

#### 4.3.2. Phytoconstituents and Preclinical Activity

Leaf, seed, stem, and also the fruit *Sechium edule* are rich in various bioactive components, as well as flavonoids, phenolics, vitamin C, and carotenoids [94,95,98]. The leaves contained the highest concentration of luteolin glycosides, while the most significant concentration of apigenin derivatives (C-glycosidic and O-glycosidic bonds) was found in the root extract [92]. Trans-cinnamic acid, phenylacetic acid, and α-linolenic acid were identified in the leaf extract [99]. Fruits can contain bitter principles called cucurbitacins [86,100,101].

In isolated aorta rings without endothelium, a hydro-alcoholic root extract caused a concentration-dependent vasorelaxation of angiotensin II-induced vasocontraction [91]. Furthermore, the authors reported in vivo antihypertensive effects in mice treated with angiotensin II [91]. The distinctive components of the highest active fraction were identified as cinnamic compounds, such as cinnamic acid methyl ester [91,97]. An aqueous leaf extract administered at 200 mg/kg showed nephroprotective activity against various types of chemically induced renal damage in rats [102]. The extract used at 100–200 mg/kg showed anti-inflammatory activity reducing levels of TGF-β, TNF-α, and ICAM-1 [103,104]. In rats fed with a high-fat diet, *Sechium edule* shoots can prevent hepatic steatosis and attenuate fatty tissue by inhibiting lipogenic enzymes and stimulating lipolysis by upregulating AMP-activating protein kinase (AMPK) [105]. The same authors also showed that the shoot extracts inhibited the expressions of fatty acid synthase and HMG-CoA reductase in rats, while also isolated caffeic acid and hesperetin, the main characteristic components of *Sechium edule* shoots, prevented hepatic lipid accumulation [105]. The acetone fraction of the hydro-alcoholic extract of *Sechium edule* roots administered to female mice at 10 mg/kg per day, orally, for 10 weeks was able to control hypertension, as well as the oxidative and inflammatory status in the kidneys, as efficiently as losartan, returning mice to normotensive levels [97,103]. Furthermore, the acetonic fraction was more effective than losartan in preventing liver and kidney damage. Therefore, the fraction was able to control endothelial dysfunction and related diseases [97,103].

#### 4.3.3. Therapeutic Efficacy: Clinical Trials

As far as was found in the literature, no clinical studies have been conducted in patients using this plant as a single treatment. A clinical trial studied a commercial antioxidant supplement containing three components, including *Sechium edule*, showing an improvement of the hemorheology in alcoholics [106]. Based on available data, clinical studies are required on the use of *Sechium edule* in hypertension, diabetes mellitus, obesity, and, in general, in vascural-related diseases.

#### 4.3.4. Safety

Acute toxicity was tested in rats and mice that received a single oral dose of 2000 mg/kg of an aqueous leaf extract. Treated animals showed no change in the normal behavior pattern and no evidence of toxicity and mortality [102]. Negative effects on humans were not documented. In VigiBase^®^ there are no reports of potential side effects related to *Sechium edule* [40], and the literature does not provide data on the safety of medicinal use during pregnancy or breastfeeding.

#### 4.3.5. Future Needs

Although traditional use and preclinical data suggest great interest in the use of *Sechium edule* in vascular diseases, the total absence of clinical studies strongly limits its use. Therefore, proper clinical trials are desirable.

### 4.4. Terminalia chebula Retz.

#### 4.4.1. Botanical Characteristics

*Terminalia chebula* (Combretaceae), also known as black myrobalan, is a deciduous tree that grows up to 30 m, widely known in India and Southern Asia for its use in Ayurvedic medicine [107,108]. Fruits are used in traditional medicine to treat various diseases, such as used as a laxative, stomachic, tonic, and antispasmodic [107,108].

#### 4.4.2. Phytoconsituents and Preclinical Activity

The main components of the fruit are phenolic compounds, such as hydrolysable tannins and flavonoids, saccharides, such as D-glucose, D-fructose, and saccharose [67]. The aqueous extract of *Terminalia chebula* fruits contains chebulagic acid, chebulinic acid, and other low molecular weight hydrolysable tannins [68]. The leaves contain polyphenols such as punicalin, punicalagin, terflavins B, C, and D [108].

The antioxidant activity of *Terminalia chebula* has been reported in vivo and in vitro assays [108,109,110,111]. A study described a significant decrease in glucose level in normal and alloxan-induced diabetic rats four hours after oral administration of a methanolic fruit extract (100 mg/kg) [110]. Similarly, *Terminalia chebula* fruit extract at a concentration of 200 mg/kg administered for 30 days significantly reduced blood glucose, glycosylated hemoglobin, urea, and creatinine levels in diabetic rats [112]. Furthermore, a cardioprotective effect of an ethanolic extract of fruits (500 mg/kg) was described in rats [113].

#### 4.4.3. Therapeutic Efficacy: Clinical Trials

Few trials have been reported in the literature on the potential therapeutic use of *Terminalia chebula*. A 12-week prospective trial showed that an aqueous extract of *Terminalia chebula* fruits administered at 250 mg and 500 mg, twice daily, significantly improved endothelial function, systemic inflammation, and lipid profile in 60 subjects with type 2 diabetes mellitus of either gender, compared to placebo treatment [66]. *Terminalia chebula* extract significantly increased NO^•^ and GSH levels, reducing oxidative stress, malondialdehyde, and CRP levels [66]. Previously, the same authors reported similar beneficial effects in 56 patients of either gender with metabolic syndrome [68]. In this study, *Terminalia chebula* reduced malondialdehyde levels and increased glutathione levels, improving antioxidant status. Furthermore, the treatment significantly decreased total cholesterol, triglycerides, and low-density lipoprotein cholesterol, and increased high-density lipoprotein cholesterol, while the placebo did not have a significant effect on endothelial function or any of the other clinical parameters [68].

#### 4.4.4. Safety

Treatment with *Terminalia chebula* extracts was well tolerated with very few side effects; in fact, very few patients experienced dyspepsia [66]. Other authors reported increased libido, dry mouth, colic, and confusion [114]. In general, the fruit preparations are well-tolerated and do not adversely affect health [67,108]. In VigiBase^®^, *Terminalia chebula* has five reports of potential side effects, such as gastrointestinal disorders (i.e., diarrhea and gastrointestinal pain), and other general disorders [40].

#### 4.4.5. Future Needs

The few clinical studies that have evaluated the efficacy of *Terminalia chebula* fruit extracts suggest a potential use in the treatment of vascular dysfunction in diabetes mellitus and/or metabolic syndrome. Unfortunately, data are limited, and other studies are certainly necessary to determine the efficacy and safety of this medicinal plant.

### 4.5. Plant-Derived Nanovesicles

#### 4.5.1. General Characteristics

In recent years, various investigations have suggested that plant cells, through an exosome-like process, may release nanosized particles, which are involved in plant cell-cell communication [115,116]. Furthermore, various studies suggest that plant-derived nanovesicles may also play a role in the properties of medicinal plants in human diseases, mainly based on their biological cargo [117,118,119,120]. It is assumed that the interaction between vegetal extracellular vesicles and mammalian cells may have beneficial effects through antioxidant and anti-inflammatory activities [121,122]. Plants produce nanovesicles in response to numerous biotic and abiotic environmental stresses, including pathogen infection and attack. Plant nanovesicles carry a wide variety of molecules, including proteins, lipids, miRNAs, vitamins, and various plant metabolites [123].

#### 4.5.2. In Vitro and In Vivo Studies

Lemon and strawberry-derived nanovesicles showed antioxidant activity in mesenchymal stem cells [124,125]. The authors verified the potential anti-osteoporotic effects of apple-derived nanovesicles using MC3T3-E1 cells, inhibiting osteoporosis by promoting osteoblastogenesis in osteoblastic MC3T3-E1 cells, by regulating the BMP2/Smad1 pathway [126]. Lemon-derived nanovesicles have been found to be rich in citric acid and vitamin C, which have a significant protective effect on oxidative stress in mesenchymal stromal cells [124].

Few studies have been carried out in animal models and humans. Oral administration of grape exosome-like nanoparticles showed beneficial effects in dextran sulfate sodium (DSS)-induced experimental colitis in mice, via induction of intestinal stem cells [120]. Similarly, broccoli-derived nanoparticles administered orally protected against various types of mice colitis, through activation of adenosine monophosphate-activated protein kinase (AMPK) in dendritic cells [127]. Additionally, ginger-derived nanoparticles protected mice from alcohol-induced liver injury, activating nuclear factor erythroid 2-related factor 2 (Nrf2) and inhibiting ROS production [128].

#### 4.5.3. Therapeutic Efficacy: Clinical Trials

Very few trials considered the administration of plant-derived nanovesicles in the treatment of human diseases. In a prospective open-label study, 20 healthy volunteers (9 women and 11 men) were treated with a commercial preparation of extracellular vesicles from *Citrus limon* L., administered at 1000 mg daily for 3 months [69]. A decrease in waist circumference was found in women after 4 and 12 weeks of treatment, while no significant reduction was detected in men [69]. In the same study, the authors also observed a significant reduction in low-density lipoproteins (LDL) [69]. Significant correlations were also found in the stratified analysis between alkaline phosphatase enzymes (ALP) and glucose for women and between ALP and LDL for men [69]. A phase 1 clinical trial is currently underway studying the ability of a grape exosome preparation, administered orally for 35 days, to act as an anti-inflammatory agent against oral mucositis during radiation and chemotherapy treatment for head and neck tumors (NCT01668849) [129]. Another clinical study with ginger and aloe-derived exosomes studying the ability to mitigate insulin resistance and chronic inflammation in patients diagnosed with polycystic ovary syndrome was withdrawn because the investigator left the university before study approval (NCT03493984).

#### 4.5.4. Safety

As reported in the literature, there are no toxicity studies or reports of undesirable effects related to human administration of these types of preparations. The use of plant-derived extracellular vesicles represents a new and very interesting approach in the treatment of diseases; however, other studies are needed to explore the advantages and, also, the disadvantages of plant-derived nanovesicles in therapy [129].

#### 4.5.5. Future Needs

Plant-derived nanovesicles are certainly an innovative type of plant preparation, which also has considerable industrial implications. However, there are many aspects to be validated, starting from the techniques for obtaining and the definition of the constituents, up to the possible uses in the prevention or treatment of human pathologies.

## 5. Conclusions

Based on the data collected, it can be observed that clinical studies concerning the use of products of plant origin in the treatment of human pathologies and, in particular, in cardiovascular diseases are few and consider only small groups of subjects. Furthermore, the studies generally do not examine the differences in treatment response comparing the female or male gender. In studies in which the efficacy of the products used were reported separately, women versus men, it was not possible to obtain evidence of the difference in efficacy because the number of subjects enrolled in the trials was too small to perform any statistical estimation.

Among the plants considered, garlic has been the most studied and there are several data on its effectiveness in the treatment of vascular-related disorders. However, the available data are insufficient to validate the pharmacological use of garlic preparations for any of the conditions under consideration. Additional research that recruits more patients is desirable. Some plants, such as *Campomanesia xanthocarpa*, *Sechium edule*, and *Terminalia chebula*, have been proposed for their potential use in vascular problems in diabetic or hypertensive subjects. Finally, a new type of innovative preparation based on plant-derived extracellular vesicles has been suggested, but this is only an idea that still requires long investigation. Importantly, greater attention must be paid in carrying out clinical trials with the aim of obtaining a personalized use of plant products, noting the differences in the effectiveness between women and men. Improved consideration of gender-based medicine is required to improve the efficacy of therapeutic interventions and reduce adverse reactions.

## Figures and Tables

**Figure 1 life-13-00866-f001:**
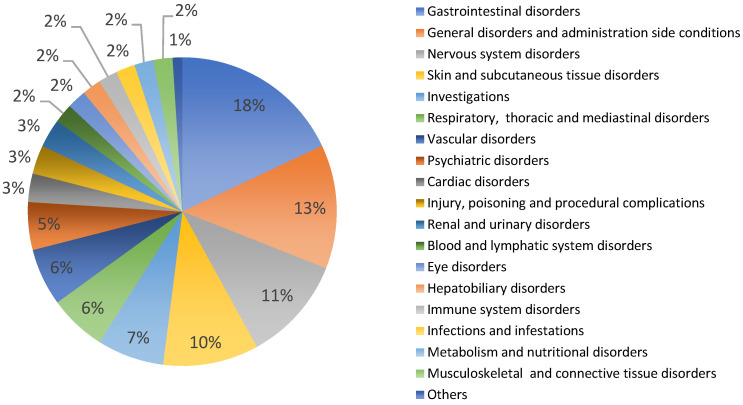
*Allium sativum*: potential side effects reported in VigiBasis^®^.

## Data Availability

Not applicable.

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
