# Peer review of "The Use of Medicinal Plants in Blood Vessel Diseases: The Influence of Gender"

_life, 2023, doi:10.3390/life13040866_

Round 1
Reviewer 1 Report
The present article entitled “The use of medicinal plants in blood vessel diseases: the influence of gender” has an interesting topic.
However, some information is misleading or not complete.
The introduction part is to small and the information is truncated. More information needs to be added so that the reader can better understand the ideas.
Line 22: please give examples of pro inflammatory cytokines
Line 72: life style risk factors
Line 82: lower coronary artery and abdominal aorta calcifications ?
Line 180: Please explain why garlic preparations is not indicated during pregnancy and lactation.
Please write for the following plant products discussed whether they can be recommended during pregnancy or not. If not, why ? (ex: limited data, no data… etc)
In the end, if clinical studies for plant products are few and they do not disclose the gender differences regarding treatment efficacy/response, the title should be changed/modified, in my opinion
Author Response
The author thanks the reviewer for comments and suggestions that helped improve the manuscript. Thus, the introduction was implemented to better explain and support the reader in understanding the ideas of the review.
The introduction was implemented adding more information about regarding TNFα role and vascular inflammation Section. The inflammatory process is particularly complex and has multiple marker modulation. The main mechanisms have been described, but not systematically, because that was not the specific objective of this review. Emphasis was placed on the evaluation of plant products used in vascular diseases, also pointing out gender differences in effectiveness when the data were available in the literature.
- Line 22: Examples of cytokines were added.
- Line 72: It has been modified as proposed.
- Line 82: It has been modified as proposed.
- Line 180: Thank you for your helpful comment. There are not sufficient studies on the safety of the medical use of garlic during pregnancy and breastfeeding. Therefore, in the absence of adequate evidence on the usefulness and safety of garlic preparations, it is recommended to avoid unnecessary use of products that could cause harm to the foetus or infant. The manuscript text was implemented by including a bibliographic reference.
Unfortunately, data relating to the use of medicinal plants in pregnancy are very limited, some regarding garlic preparations, and therefore it is not possible to define the risk/benefit ratio in pregnancy or during lactation. To achieve a final evaluation, more clinical trials are required.
It is true that clinical studies for plant products are few and do not disclose gender differences in treatment efficacy/response, but the title is a tool to highlight the value of studying efficacy and safety in both sexes. For this, I think it is important to maintain the proposed title in the manuscript to push researchers in future clinical trials to focus on the gender difference in efficacy/toxicity of herbal products (and also with respect to all types of drugs). Furthermore, in the manuscript, when reviewing the literature, plant-derived products were selected for which both male and female subjects were enrolled, as shown in Figure S1 and in the methods, in order to detect potential differences in the two-sex responses.
The authors are grateful to the reviewer for the recommendations that helped improve the manuscript.

Reviewer 2 Report
The present review analyzes the effect of four plants on vascular inflammatory diseases reported in the last ten years. It analyzes basic experimental and clinical studies. However, the author's main focus is on human studies. The structure of the review makes it very easy to understand the subject. It indicates both the compounds that have been found in plants and the biological effects in humans and in the case of chayote the effects in rodents.
Author Response
The author thanks the reviewer for the evaluation of the article and the positive assessment.

Reviewer 3 Report
This review presented by Guglielmina Froldi aimed to analyze the Data available in the literature regarding the use of herbal products to treat and prevent inflammation-related vascular diseases with a special focus on the influence of gender. Based on Pubmed and VigiBase searching strategy, the author has selected, for a bibliographic analysis, some plants (Allium sativum, Campomanesia xanthocarpa, Sechium edule, and Terminalia chebula, preparation consisting of plant-derived nanovesicles) of clinical interest for the treatment and/or prevention of vascular diseases. The paper is well written and well referenced (124 references) and all chapters are presented in a logical way that contain recent information (>35% over the past 5 years). In the introduction section, the author provides an overview on the inflammation process, its biological markers and some variation factors including the sex, to make the reader better understand the other chapters of the manuscript.
In my opinion the manuscript is suitable for publication, However the author is invited to comment on the following points:
1- The various chapters developed in the bibliographic anlaysis of each plant are very relevant however the critical anlaysis of each part is missing by the author who chose to make a very brief discussion in the conclusion. It would have been useful for the author to make an analysis of the benefit/risk ratio of each plant at the end of each chapter and not just to describe the data of the literature
2- It is questionable as to whether the Pubmed and VigiBase searching strategy is sufficiently relevant to draw conclusions as stated in the paper. The author must comment on the limitations of the findings (data analysis of the selected plants) of the manuscript due to the fact that literature analysis was based mainly only on 2 databases.
3- it would be useful for the author to clarify to the reader the exact nature of the plant tested in the clinical studies indicated in the manuscript (dietary supplement or botanical, or others…). This point should be addressed in the manuscript.
Author Response
The author thanks the reviewer for evaluating the manuscript and the positive assessment.
- Thank you for the suggestion. At the end of each medicinal plant report, a further section has been added entitled ‘Future needs’ describing the benefit/risk ratio in relation to the available data in the literature.
- This research began with a search in the PubMed database, a well-known open source resource containing more than 35 million citations and abstracts from the biomedical literature. Successively, the selected articles were read and various references in them were also evaluated. Google Scholar was also interrogated. Safety data was extrapolated from the reviewed literature and also using VigiBase, the WHO's unique global database of reported potential side effects of medicinal products (https://who-umc.org/vigibase/). Therefore, although the research carried out may not contain all the data available in the scientific literature, I hope that it can be considered exhaustive and interesting for readers interested in the field of medicinal plants.
- The types of plant extracts tested in the clinical trials were added in the manuscript, unless the information is not available in the published study. However, no brand names have been added to prevent advertising of products that may be marketed.
I would like to thank the reviewer for his/her help in improving the manuscript.

Round 2
Reviewer 1 Report
Overall, all my requirements have been adressed. The article has been improved, therefore I recommend its publication.
Author Response
The author thanks the academic reviewer for comments and suggestions that certainly helped to improve the manuscript.
Reviewer. I would like to provide you with some recommendations before proceeding with the process. In general, I believe that the title of the manuscript does not reflect the content that can be found in both the abstract and the introduction.
Answer. In relation to this point, some changes, also following the suggestions of the reviewer in detail below, were introduced in both the abstract and the introduction to try to highlight that one of the purposes of this manuscript was to examine the differences in response to plant products in relation to gender. This may make the title of the manuscript more appropriate.
Reviewer question. What is the current state of studies measuring the influence of gender on the effect of medicinal plants?
Answer. Unfortunately, information on the diversity in response to treatment with herbal medicines in women compared to men is almost non-existent, as this research demonstrates.
Reviewer question. What led the author to select the chosen medicinal plants?
Answer. The medicinal plants reported in the review were selected based on research conducted in PubMed, as described in the Methods section.
Reviewer question. Why has the review focused on this specific topic?
Answer. The present review focused on this point, first, to examine existing data on the difference between genders in the efficacy of medicinal plants and secondly to stimulate researchers to investigate this matter in the next clinical trials. Gender-specific analyses must be conducted and published for both efficacy and safety.
Reviewer question. What are the unresolved needs?
Answer. To address this point and, in general, the lack of scientific evidence on the use of medicinal plants, appropriate clinical studies are necessary that involve a sample of individuals stratified by sex.
- As suggested, the introduction was improved by merging Paragraphs 1, 2, and 4. Furthermore, a section related to methods was added. Additionally, the Results section was added. However, point 3, reporting the sex differences in vascular function, was left after the first part of the introduction and not in the section Results because it was not obtained from the PubMed search.
Thank you.
